

# Standardization of *in situ* coral bleaching measurements highlights the variability in responses across genera, morphologies, and regions

Adi Khen[1], Christopher B. Wall[2] and Jennifer E. Smith[1]

[1] Center for Marine Biodiversity and Conservation, Scripps Institution of Oceanography, University of California San Diego, La Jolla, CA, United States of America
[2] Division of Biological Sciences, University of California San Diego, La Jolla, CA, United States of America

## ABSTRACT

Marine heatwaves and regional coral bleaching events have become more frequent and severe across the world's oceans over the last several decades due to global climate change. Observational studies have documented spatiotemporal variation in the responses of reef-building corals to thermal stress within and among taxa across geographic scales. Although many tools exist for predicting, detecting, and quantifying coral bleaching, it remains difficult to compare bleaching severity (*e.g.*, percent cover of bleached surface areas) among studies and across species or regions. For this review, we compiled over 2,100 *in situ* coral bleaching observations representing 87 reef-building coral genera and 250 species of common morphological groups from a total of 74 peer-reviewed scientific articles, encompassing three broad geographic regions (Atlantic, Indian, and Pacific Oceans). While bleaching severity was found to vary by region, genus, and morphology, we found that both genera and morphologies responded differently to thermal stress across regions. These patterns were complicated by (i) inconsistent methods and response metrics across studies; (ii) differing ecological scales of observations (*i.e.*, individual colony-level vs. population or community-level); and (iii) temporal variability in surveys with respect to the onset of thermal stress and the chronology of bleaching episodes. To improve cross-study comparisons, we recommend that future surveys prioritize measuring bleaching in the same individual coral colonies over time and incorporate the severity and timing of warming into their analyses. By reevaluating and standardizing the ways in which coral bleaching is quantified, researchers will be able to track responses to marine heatwaves with increased rigor, precision, and accuracy.

# INTRODUCTION

## Reviewing the causes and consequences of coral bleaching

Reef-building corals, the ecosystem engineers for tropical coral reefs, exist in a mutualistic symbiosis with photosynthetic dinoflagellate symbionts (Symbiodiniaceae; *LaJeunesse et al., 2018*) that support coral nutrition and growth (*Muscatine & Porter, 1977*). However,

Corresponding author
Adi Khen, akhen@ucsd.edu

environmental stress—notably, marine heatwaves—can push this symbiosis into a state of dysbiosis, with the coral losing its symbionts in a process termed "coral bleaching". Bleaching increases a coral's vulnerability to complete or partial mortality, disease, and colony fragmentation, and can reduce coral growth and/or reproduction (*e.g.*, *Baird & Marshall, 2002*; *Brown, 1997*; *Buddemeier, Kleypas & Aronson, 2004*; *Hoegh-Guldberg, 1999*; *Jokiel & Brown, 2004*). This can be exacerbated by prolonged duration (weeks to months) and/or increased magnitude of thermal stress (*Cook et al., 1990*). However, corals can recover from bleaching once non-stressful conditions are restored (*Jones & Yellowlees, 1997*). During this period of dysbiosis and post-stress recovery, corals can compensate for the lack of symbiont-derived nutrition by feeding heterotrophically on suspended particles and plankton (*Fox et al., 2019*; *Grottoli, Rodrigues & Palardy, 2006*; *Palardy, Rodrigues & Grottoli, 2008*) or relying on the consumption of energy reserves, such as lipids, to sustain metabolism (*Grottoli, Rodrigues & Juarez, 2004*; *Porter et al., 1989*; *Rodrigues & Grottoli, 2007*; *Wall et al., 2019*). Ultimately, corals surviving bleaching events may undergo shifts in their endosymbiont community assemblages to promote thermal tolerance (*e.g.*, *Kemp et al., 2014*; *Jones et al., 2008*). To better understand the dynamics of coral bleaching and recovery—as well as the individual, local, or regional factors contributing to bleaching susceptibility or tolerance—there is a need for more precise colony-level data incorporating the severity of bleaching responses and the trajectory of bleaching recovery in relation to local and regional environmental conditions.

Many environmental triggers aside from warming can result in (or exacerbate) coral bleaching, including reduced salinity (*e.g.*, *Goreau, 1964*; *Van Woesik, De Vantier & Glazebrook, 1995*), solar radiation (*e.g.*, *Brown et al., 1994*; *Lesser et al., 1990*), and bacterial infection (*e.g.*, *Brown, 1997*; *Kushmaro et al., 1996*). However, widespread coral bleaching events are largely due to marine heatwaves that cause anomalously high seawater temperatures and are driven by global climate change (*Heron et al., 2016*; *Hughes et al., 2017a*; *Spalding & Brown, 2015*). Mass coral bleaching events (*i.e.,* multiple geographic locations experiencing bleaching simultaneously) occur in areas of high accumulated thermal stress, where sea surface temperatures (SST) have exceeded the local bleaching threshold (*i.e.,* one degree Celsius above maximum monthly mean) for multiple consecutive weeks. Mass coral bleaching was first described in scientific literature in 1984, following the severe El Niño-Southern Oscillation (ENSO) event from 1982–1983 (*Glynn, 1984*). As of 2011, regional bleaching has been documented over 7,000 independent times worldwide (ReefBase; *Donner, Rickbeil & Heron, 2017*). In the span of the past few decades, bleaching has been reported in nearly every location where coral reefs exist across the globe (Fig. 1, data obtained from ReefBase and *Donner, Rickbeil & Heron, 2017*). In an effort to catalog these bleaching events and their consequences, a historical coral bleaching database has been compiled (https://simondonner.com/bleachingdatabase/) which is currently the most comprehensive archive of bleaching records publicly available. This archive combines observations from the non-profit global information system, ReefBase (http://www.reefbase.org), with reports by researchers and reef managers. For instance, the Great Barrier Reef predominantly suffered from a regional coral bleaching event in 2002 while the 2005 event was centralized on the Caribbean (Fig. 2, data obtained from ReefBase

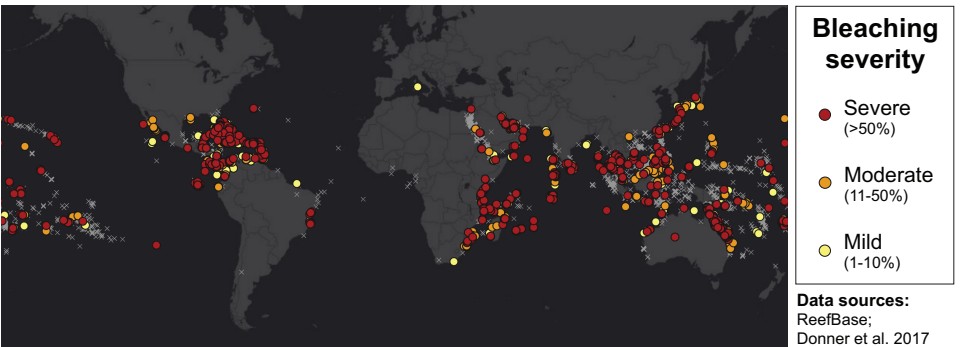

**Figure 1** **Mapping all areas historically affected by mass coral bleaching.** World map showing the location and severity of mass coral bleaching events from 1963 to 2011 (*Donner, Rickbeil & Heron, 2017*). Colored circles indicate bleaching severity and light gray crosses indicate documented locations where coral reefs exist (ReefBase). Basemap source: Esri, DigitalGlobe, GeoEye, Earthstar Geographics, CNES/Airbus DS, USDA, USGS, AeroGRID, IGN, and the GIS User Community.

and *Donner, Rickbeil & Heron, 2017*). To date, three global coral bleaching events were observed across all main tropical ocean basins in 1998, 2010, and 2015 (*Eakin, Sweatman & Brainard, 2019*; *Heron et al., 2016*). Since 1980, widespread bleaching has occurred most frequently in the western Atlantic, followed by the Indian and Pacific Ocean basins (*Hughes et al., 2018*). The 2015 global coral bleaching event lasted for an unprecedented three successive years of bleaching (2014–2017; *Eakin, Sweatman & Brainard, 2019*) in some locations and hit nearly every major tropical region on Earth (Fig. 3, data obtained from NOAA's Coral Reef Watch Program). Altogether, this has allowed for geographic explorations of patterns in bleaching prevalence.

## Bleaching resistance and resilience at physiological and ecosystem scales

Resistance to thermal stress can be defined as the ability of individual corals to avoid bleaching or survive post-bleaching (*West & Salm, 2003*). It is generally accepted that corals have different susceptibilities to bleaching based on taxonomy (*Marshall & Baird, 2000*; *McClanahan et al., 2004*), life history strategy (*Darling, McClanahan & Côté, 2013*), morphology (*Loya et al., 2001*; *Van Woesik et al. 2012*), colony size class (*Shenkar, Fine & Loya, 2005*), symbiont type and/or density (*Berkelmans & Van Oppen 2006*), as well as any other distinguishing characteristics. 'Weedy' genera (*i.e.,* taxa that are fast-growing, opportunistic, and able to dominate post-disturbance, such as *Acropora* or *Pocillopora* spp.) may be more sensitive to bleaching yet quicker to regain their pre-bleaching benthic cover following substantial colony mortality (*Darling, McClanahan & Côté, 2013*; *McClanahan, Graham & Darling, 2014*).  Morphology is also thought to play a role, though results are often contradictory. For instance, mounding corals show high (*Marshall & Baird, 2000*) or low (*Williams et al., 2010*) resistance to thermal bleaching. Branching corals may experience more bleaching-related mortality than massive or encrusting corals (*Hoegh-Guldberg & Salvat, 1995*; *Marshall & Baird, 2000*), presumably because the thinner tissues of branching
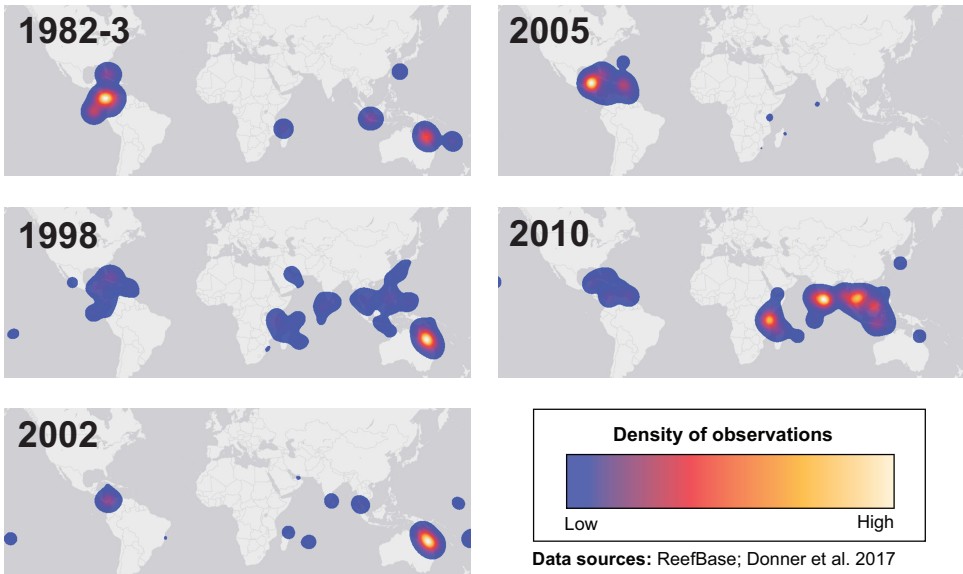

**Figure 2  Mapping spatial distributions of mass coral bleaching events.** Heatmaps showing the spatial distributions of each major bleaching event between 1982 and 2010. Basemap source: Esri, DeLorme, HERE, MapmyIndia.

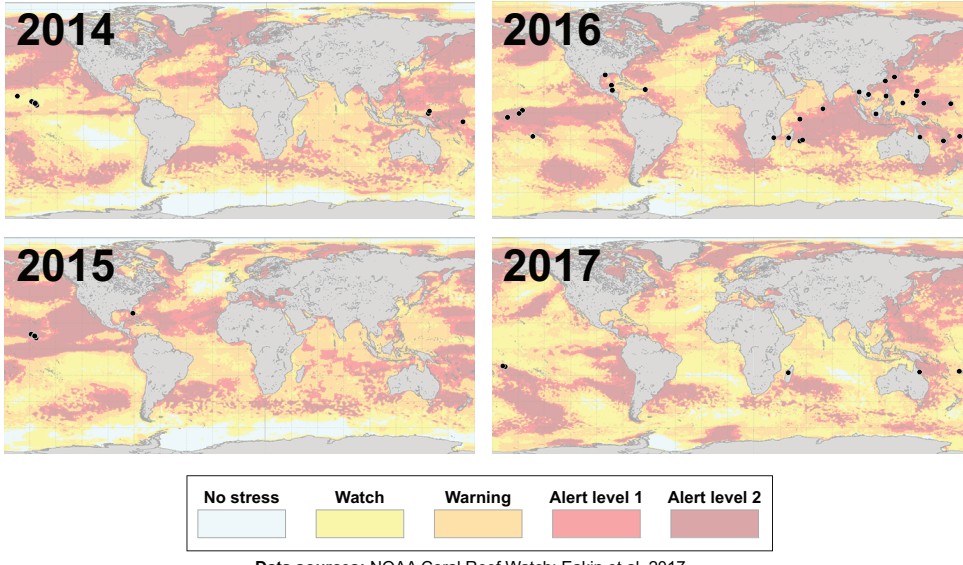

**Figure 3  Mapping the impact of the third-ever global coral bleaching event.** Maps showing over 75 total locations worldwide, represented by black circles, affected by the third-ever global coral bleaching event lasting between 2014 and 2017 (*Eakin, Sweatman & Brainard, 2019*). Each map is overlaid with maximum heat stress in that respective year from NOAA's Coral Reef Watch satellite data. Basemap source: NOAA Coral Reef Watch.

corals expose symbionts to higher light intensities (*Loya et al., 2001*). Larger colonies with more tissue area could hypothetically have an advantage over smaller colonies during bleaching due to their higher symbiont densities, although some studies indicate otherwise (*Brandt, 2009*; *Shenkar, Fine & Loya, 2005*; *Wagner, Kramer & Van Woesik, 2010*). Tissue biomass may also correlate with bleaching resistance and high symbiont density, as species with lower tissue biomass have been found to experience increased mortality following bleaching (*Thornhill et al., 2011*).

Additionally, it has long been posited that certain Symbiodiniaceae assemblages may be more stress-tolerant and that colonies may change the relative abundance of different co-occurring symbionts in their tissues, thereby shifting dominance (or proportions) of the symbiont community following bleaching (*Kemp et al., 2014*; *Mieog et al., 2007*; *Baker, 2003*; *Toller, Rowan & Knowlton, 2001*). Studies have since found a high degree of specificity among coral-Symbiodiniaceae associations in the long term (*Lee et al., 2016*; *Stat et al., 2009*; *Thornhill, Fitt & Schmidt, 2006*), implying that the uptake of new symbionts with different stress tolerance may be transient (*Coffroth et al., 2010*), particularly for adult corals (but see: *Scharfenstein et al., 2022*; *Boulotte et al., 2016*). Further, some coral taxa exhibit higher flexibility in their Symbiodiniaceae assemblages than others (*Goulet, 2006*; *Putnam et al., 2012*) and individual colonies may have their own 'Symbiodiniaceae signature' (*Rouzé et al., 2019*). The dynamics of these symbioses are also affected by environmental regimes (*Baker et al., 2013*; *de Souza et al. 2023*; *de Souza et al. 2022*) and the symbionts' physiological traits (*Wong, Enríquez & Baker, 2021*), making it difficult to predict post-bleaching recovery or mortality based solely on changes in the composition of *in hospite* symbiont communities.

Physical factors such as location (*e.g.*, *Sully et al., 2019*), reef habitat type (*e.g.*, *Wagner, Kramer & Van Woesik, 2010*; *Wall et al., 2021*), human population density (*e.g.*, *Sandin et al. 2008*), and/or environmental variability (*e.g.*, *Bahr, Rodgers & Jokiel, 2017*; *Jokiel & Brown, 2004*) may also affect bleaching outcomes, along with prior bleaching history (*e.g.*, *Brown et al., 2002*; *Pratchett et al., 2013*), maximum heat stress (*Claar et al., 2018*), and local disturbance (*Baum et al., 2023*). Depth is thought to provide a refuge for corals from solar heating and light (*Baird et al., 2018*; *Smith et al., 2014*) as well as driving niche partitioning of functionally distinct Symbiodiniaceae communities (*Wall et al., 2020*). However, depth does not always confer resistance to bleaching, as has been seen in both the Caribbean (*Neal et al., 2014*) and Pacific (*Venegas et al., 2019*). Factors that reduce thermal stress (*e.g.*, cold-water upwelling; *Goreau et al., 2000*), enhance water flow and flush out cytotoxic ions (*Nakamura & Van Woesik, 2001*), and decrease light stress (*e.g.*, shading from cloud cover (*Mumby et al., 2001*)) or light absorption by dissolved organic matter (*Anderson et al., 2001*) can determine bleaching resistance.

While coral physiological resistance pertains to the ability to withstand or not be harmed by a disturbance, the physiological resilience of corals and the ecological resilience of coral reefs is the ability for the coral holobiont or the reef community to recover from or return to pre-disturbance physiological or ecological states, respectively (*West & Salm, 2003*). The speed at which this occurs, along with the magnitude of disturbance, can also be considered when defining resilience (*Gunderson, 2000*; *Nyström, Folke & Moberg, 2000*). Ecosystem

resilience is determined by either intrinsic (*e.g.*, larval production capacity and recruitment success, or the presence of herbivorous grazers (*Heenan & Williams, 2013*; *Roff & Mumby, 2012*)) or extrinsic (*e.g.*, effective management and protection (*Mellin et al., 2016*; *Salm & Coles, 2001*)) factors. Some remote, protected reefs such as the Phoenix Islands in the central Pacific have shown evidence of increased resilience following successive heatwaves, possibly due to adaptive thermal tolerance and localized recruitment by surviving colonies (*Fox et al., 2021*). However, the 2016 heatwave also caused severe (>50%) mortality throughout the most remote sections of the Great Barrier Reef (*Hughes et al., 2017b*) and in protected areas of the Northwestern Hawaiian Islands (*Couch et al., 2017*), where the human impacts that may further exacerbate thermal stress (*e.g.*, urbanization, nutrient pollution) or degrade reef ecosystem function (*e.g.*, overfishing) are largely absent.

Thus, in the context of global climate change, there are perhaps no refugia where corals are not threatened by marine heatwaves and regional bleaching events. By the year 2050, bleaching is predicted to occur annually for all reefs globally (*Donner et al., 2005*; *Van Hooidonk et al., 2014*). By 2100, with a rise in global sea surface temperatures of about 3 °C under Representative Concentration Pathway 8.5 (*Pörtner et al., 2019*; Intergovernmental Panel on Climate Change), most reefs worldwide are projected to decrease in coral cover by over 40% (*Sully, Hodgson & van Woesik, 2022*). Post-bleaching recovery, however, might be more influenced by local stressors (*e.g.*, overfishing, pollution, sedimentation, or coastal development), the absence of which may facilitate the recovery of corals with more heat-adapted Symbiodiniaceae symbionts (see *Claar et al., 2020*) and reduce the extent of bleaching. While mitigating local stressors can potentially minimize climate impact (*Donovan et al., 2020*; but see: *Bruno, Côté & Toth, 2019*), local and global stressors could also act synergistically to magnify post-bleaching mortality (*Donovan et al., 2021*).

## Coral bleaching visual assessment methods

Current methods for visually assessing bleaching usually involve satellite remote sensing, aerial surveys, underwater surveys, or image analysis of transects or quadrats. While all of these methods contribute to our knowledge of bleaching severity, they operate under varying levels of taxonomic and spatial resolution (reviewed in *Van Woesik et al., 2022*; Fig. 4) and present their own advantages and disadvantages. Satellite remote sensing, while informative and large-scale (*e.g.*, 5 km or 50 km resolution), relies on predictions from temperature metrics rather than *in situ* bleaching data (*Liu et al., 2014*). Other complications with remote sensing include cloud cover and the fact that bleached corals can have a similar spectral signature as sand (*Elvidge et al., 2004*). Moreover, satellites can only provide bleaching forecasts whereas aerial surveys conducted *via* aircrafts (*e.g.*, *Hughes et al., 2017b*) or small unmanned drones (*e.g.*, *Levy et al., 2018*) can map entire reefs but quantify bleaching on a reef-wide basis. However, new technologies such as fluid lensing (*Chirayath & Earle, 2016*), which uses water-transmitting wavelengths to passively image underwater objects, are now being developed to improve the use of remote sensing tools and could potentially deliver centimeter-resolution data at regional scales. Similarly, airborne mapping combined with laser-guided imaging spectroscopy and deep learning models

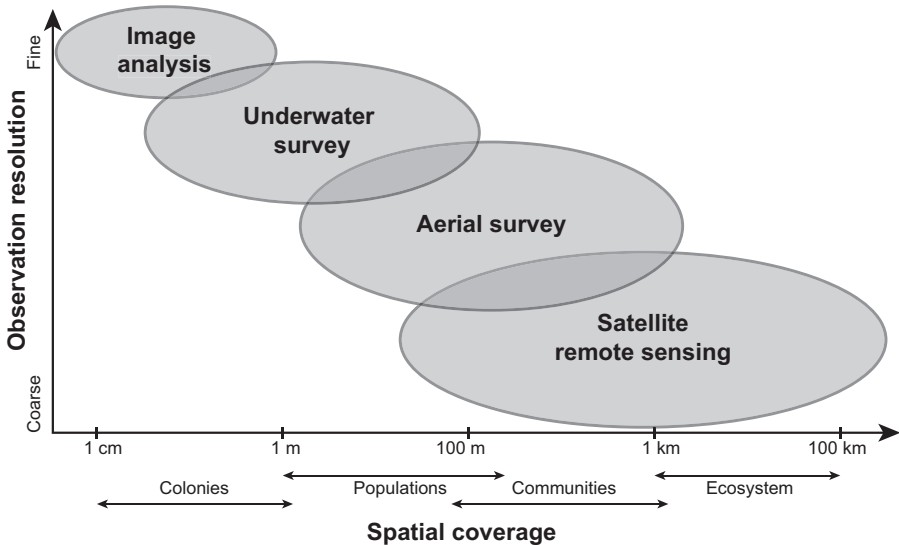

**Figure 4** **Comparing the scale and resolution of current methods for measuring coral bleaching.** Diagram comparing different methods for measuring coral bleaching in terms of their spatial coverage and observation resolution (*e.g.*, image analysis allows for the precise measurement of colony-specific bleaching but is limited spatially, whereas remote sensing is more expansive yet limited taxonomically).

(*Asner et al., 2020*) can provide regional-scale quantitative estimates of reef condition but these approaches often require optimal weather conditions.

Underwater surveys (*e.g.*, SCUBA/snorkel surveys, tow-boards) offer the opportunity to make direct observations but are more time-intensive and vary in scale from focal colony assessments to towed-diver surveys at the community or reefscape level. Alternative approaches for surveying entire reef communities are diver-led surveys at the tens-of-meters scale (*i.e.*, point-intercept and transects), which can give insights into population or colony-level bleaching responses. Corals are often categorized as "bleached", "unbleached", or another qualitative bleaching group (*McClanahan, 2004*) rather than quantifying percent of color change relative to an unaffected (*i.e.*, non-bleached) conspecific colony, although this can be subject to observer bias (*Siebeck et al., 2006*). A color reference card (*e.g.*, *Bahr et al., 2020*; *Siebeck et al., 2006*) can also be used to visually compare coral pigmentation *in situ* to "healthy" representatives, however, these color references require local ground-truthing to capture the range of color for healthy and bleached corals in a specific region (*Bahr et al., 2020*). Image analysis, in which bleached coral planar areas within photoquadrats taken during underwater surveys are digitally traced using computer software (*e.g.*, *Neal et al., 2017*), is arguably the most precise (albeit for a more rugose coral, planar area measurements are less representative of surface areas). Still, this method can be time-consuming (*Williams et al., 2019*) and subjective, particularly in distinguishing natural paling, discoloration, or partial bleaching from thermal stress-associated bleaching. Image analysis is also traditionally small-scale at the colony level, which does not always reflect the population or community as a whole. Recent machine learning initiatives, such as exclusively-automatic bleaching detection, are now being attempted by the XL Catlin

Seaview Survey (https://www.catlinseaviewsurvey.com/). Other automated classification programs such as CoralNet (*Beijbom et al., 2015*, https://coralnet.ucsd.edu/), require extensive training data to achieve a sufficient level of taxonomic resolution and cannot measure bleaching within colonies (*Bryant et al., 2017*). In the future, a combination of human-validated, computer-annotated image analysis will likely optimize efficiency and allow for thorough examination of how corals bleach over time on an individual colony basis.

While bleaching observations across all scales have merit, in order to predict regional or taxonomic responses more accurately, it is also important to acknowledge underlying physical, biological, or contextual factors. Additionally, to gain a better understanding of trends in coral bleaching and recovery, there is a need to synthesize studies that have quantified responses from multiple scales, regions, taxa, and/or morphologies. In this paper we address the ways in which these data were collected or reported, and use a standardization approach to compare data among observational studies. Our goal was to determine how bleaching severity has varied spatially, taxonomically, and/or morphologically and identify potential patterns.

## SURVEY METHODOLOGY

### Literature search

In 2021, we conducted a literature search to identify peer-reviewed studies that focused on observational responses of corals to thermal stress (bleaching) *in situ*. We sought to determine how bleaching severity varied by (i) region and/or genus, and (ii) region and/or morphology. We used pre-defined search terms in Google Scholar and Web of Science (*e.g.*, "coral bleaching", "coral*" and "bleach*", "bleaching severity", "bleaching index", "mass bleaching") to compile existing data from studies of all-time that visually assessed coral bleaching at the population or colony level, by genus and/or species. Our literature review consisted of 74 published scientific studies; in total, there were 2,137 bleaching observations from 87 coral genera and 250 species. We then recorded the geographic location for each study, reef habitat and depth, mass bleaching event(s) experienced, time of sampling with regard to the bleaching event as noted by the authors, their method of observing or analyzing bleaching, and the study's bleaching response metric. For each individual observation, we recorded the genus and species name as noted in the original study, the current taxonomic name (updated against the World Register of Marine Species, https://www.marinespecies.org/), the general morphology or growth form for that taxon, and the quantified bleaching response according to the original study. Since coral taxa often have more than one morphology (sometimes even within the same colony), morphological classifications were not mutually exclusive both among and within studies.

### Data collection and analysis

Each study-specific bleaching observation was assigned a relative bleaching severity category (*i.e.,* from "none" to "severe") based on the quantified bleaching response reported in the study itself. When data on the percentage of individual colonies covered in bleached tissue were provided, we designated >80% bleaching as "severe", 61–80% as "high", 41–60% as

"moderate", and <40% as "low". However, since not all studies used the same response metric or rating scale (*e.g.*, bleaching prevalence—defined as the proportion of colonies affected by bleaching, or Bleaching Index—a weighted average of the relative abundance of colonies within each category of bleaching severity, developed by *McClanahan (2004)*) while some studies established a new metric *ad hoc* (*e.g.*, Cross-Correlation Coefficient as a function of coral color and seawater temperature), bleaching responses in their original form were often incomparable quantitatively. To allow for adequate comparison across studies, we assigned a bleaching severity category to each non-standard bleaching observation relative to others in that particular study. For the purpose of data visualization and analysis, these were later converted to numerical scores for each bleaching severity category as follows: 0 = none, 1 = low, 2 = moderate, 3 = high, and 4 = severe.

Of all studies that quantified coral bleaching by taxon, 10 were from the Atlantic Ocean (including the Caribbean), 21 were from the Indian Ocean (including the Red Sea), and 43 were from the Pacific Ocean. Bleaching events studied ranged from 1987 to 2017, at shallow (<5 m depth) lagoon, terrace, or reef flat habitats to deeper (>15 m depth) fore reef and reef slope habitats (File S1). Results for bleaching severity, represented by the standardized numerical severity scores, were then plotted by region (*i.e.*, each of the three main tropical ocean basins), genus, and/or morphology. Since more than half of the studies did not identify corals to species, we did not have enough information to visualize bleaching severities on a taxonomic resolution finer than the genus level.

For statistical analysis, we conducted two separate aligned rank transform analyses of variance (ART-ANOVA) with Type-II sum of squares using the *ARTool* package in R software version 4.3.0 (*Kay et al., 2021*; *R Core Team, 2018*). Our fixed factors were either region and genus, or region and morphology, and our dependent variable was standardized bleaching severity score. We also tested for interactions between each pair of factors to see whether genera or morphologies bleached differently in different regions. We used the non-parametric ART-ANOVA rather than a standard two-way ANOVA because our response data were rank-based and thus non-continuous. Post-hoc multiple comparisons were performed using the stat_compare_means function in *ggpubr* (*Kassambara, 2020*; *Wickham, Chang & Wickham, 2016*) to identify the genera or morphologies for which regions responded differently in terms of bleaching severity. A combined three-way ANOVA with region, genus, and morphology as factors was not possible since not all morphologies were represented in all genera. Also, we only used a subset of the data since not all genera or morphologies exist in all regions, which otherwise would have led to an unbalanced design.

## RESULTS

### Variability in bleaching responses by genus, morphology, and/or region

The majority of bleaching observations were from the Pacific ($n = 1,383$ observations) followed by the Indian ($n = 534$) and Atlantic ($n = 220$) Oceans. We found that bleaching severity varied significantly by region ($p = 0.001$) and genus ($p < 0.001$; Table 1). Notably,

**Table 1  Aligned rank transform analysis of variance: region and genus.** Statistical output from a two-way ART-ANOVA for the effects of region and genus on bleaching severity, as well as their interaction. Only genera present in most or all regions were included. Significant ($p < 0.05$) factors are bolded.

| Source | Df | Df.res | *F* value | Pr(>*F*) |
|---|---|---|---|---|
| **Region** | 2 | 1407 | 6.901 | **0.001** |
| **Genus** | 19 | 1407 | 3.468 | **<0.001** |
| **Region * Genus** | 22 | 1407 | 2.868 | **<0.001** |

there was a significant interaction indicating that some genera, particularly *Acropora* ($p = 0.0025$), *Favia* ($p < 0.001$), *Galaxea* ($p = 0.0095$), and *Porites* ($p = 0.003$) responded differently by region. *Acropora* experienced less bleaching in the Atlantic Ocean than in the Indian or Pacific Ocean (mean bleaching severity score $= 0.8 \pm 0.3$ standard error compared to $2.4 \pm 0.2$ and $1.6 \pm 0.1$, respectively), where it is far more abundant and speciose (*Richards, Berry & Van Oppen, 2016*). *Favia* experienced more bleaching in the Atlantic than in the Indian or Pacific Ocean (mean bleaching severity score $= 3.0 \pm 0.6$ compared to $1.4 \pm 0.2$ and $0.9 \pm 0.2$; Fig. 5). *Porites* bleached similarly in the Indian and Atlantic (mean bleaching severity score $= 1.6 \pm 0.2$ and $1.6 \pm 0.6$, respectively) but relatively less in the Pacific (mean bleaching severity score $= 0.9 \pm 0.2$). *Montastraea*, the only other genus represented across all ocean basins, did not show differences in bleaching responses by region. *Galaxea* bleached significantly more in the Indian than in the Pacific Ocean (mean bleaching severity score $= 1.8 \pm 0.2$ *vs.* $1.0 \pm 0.2$) whereas *Stylophora* was more bleached in the Pacific compared to the Indian Ocean (mean bleaching severity score $= 2.8 \pm 0.3$ *vs.* $1.9 \pm 0.4$), although the latter pattern was not found to be significant possibly due to lower sample sizes.

We also found significant effects of region ($p < 0.001$) and morphology ($p < 0.001$) on bleaching responses (Table 2). Again, there was a significant interaction between region and morphology indicating that some morphologies responded differently across regions. Specifically, massive ($p = 0.002$) and massive/encrusting ($p = 0.004$) corals bleached most in the Atlantic, followed by the Indian and Pacific Oceans (mean bleaching severity scores $= 1.9 \pm 0.1$, $1.5 \pm 0.1$, and $1.4 \pm 0.1$, or $1.9 \pm 0.2$, $1.4 \pm 0.1$, and $1.3 \pm 0.1$, respectively). In contrast, free-living corals ($p = 0.004$) bleached most in the Indian and Pacific Oceans compared to the Atlantic (mean bleaching severity scores $= 1.8 \pm 0.2$, $1.2 \pm 0.3$, and $0.2 \pm 0.1$, respectively). Branching corals showed particularly high variability in bleaching responses within regions (bleaching severity range $= 0–4$; Fig. 6) but not significantly among regions. Columnar corals ($p = 0.002$) bleached more in the Pacific compared to the Atlantic Ocean (mean bleaching severity score $= 2.0 \pm 0.3$ *vs.* $0.7 \pm 0.3$). Encrusting, plating, and table/branching corals did not bleach more severely in one region over another. In general, for both morphology and genus, there was a large spread of responses within regions, suggesting that differences could have also been related to colony-specific factors such as size, symbiont community composition, previous exposure to thermal stress, or site-level local environmental variables.
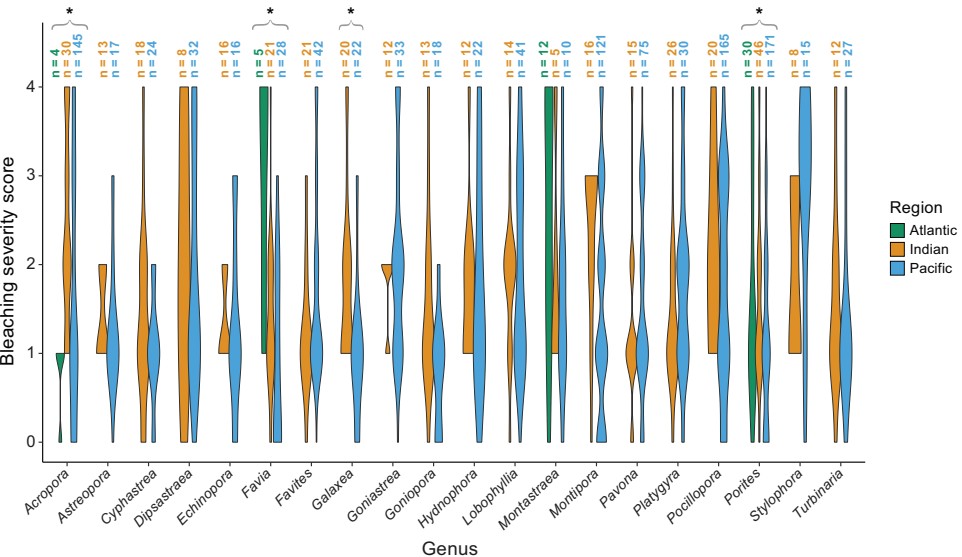

**Figure 5** **Coral bleaching severity by genus and region.** Violin plots showing the distribution of bleaching severity scores for each common coral genus across the Atlantic, Indian, and Pacific Oceans (0 = none, 1 = low, 2 = moderate, 3 = high, 4 = severe). The total number of observations by genus and region is indicated above each box. Asterisks indicate genera for which mean bleaching severity varied significantly by region.

**Table 2** **Aligned rank transform analysis of variance: region and morphology.** Statistical output from a two-way ART-ANOVA for the effects of region and morphology on bleaching severity, as well as their interaction. Only morphologies present in most or all regions were included. Significant ($p < 0.05$) factors are bolded.

| Source | Df | Df.res | *F* value | Pr(>*F*) |
|---|---|---|---|---|
| **Region** | 2 | 1792 | 16.923 | **<0.001** |
| **Morphology** | 7 | 1792 | 7.370 | **<0.001** |
| **Region * Morphology** | 13 | 1792 | 5.112 | **<0.001** |

## Variability in bleaching quantification among studies

A total of 68.9% of studies (51 out of 74; Table S1) assessed bleaching *via* rapid *in situ* surveys either by snorkelers, SCUBA divers, or tow-board; whereas 18.9% of studies (14 out of 74; Table S1) used exclusively image analysis methods, either on a small scale (*e.g.*, quadrats) or larger scale (*e.g.*, transects, mosaics, or photostations). 5.4% of studies (4 out of 74; Table S1) used exclusively video analysis, while the remaining 6.8% used a combination of *in situ* surveys and image or video analysis methods.

Bleaching response metrics were widely inconsistent across studies (Table S2). For 62.2% of studies (46 out of 74; Table S2), colonies were categorized as either completely bleached, mostly, somewhat, or slightly bleached, pale or mottled, affected by bleaching, "healthy", or other *ad hoc* categories as denoted by authors. These studies used the proportion of colonies that fell within each category to quantify bleaching prevalence for a given genus and/or species. A similar approach used by 21.6% of studies (16 out of 74;

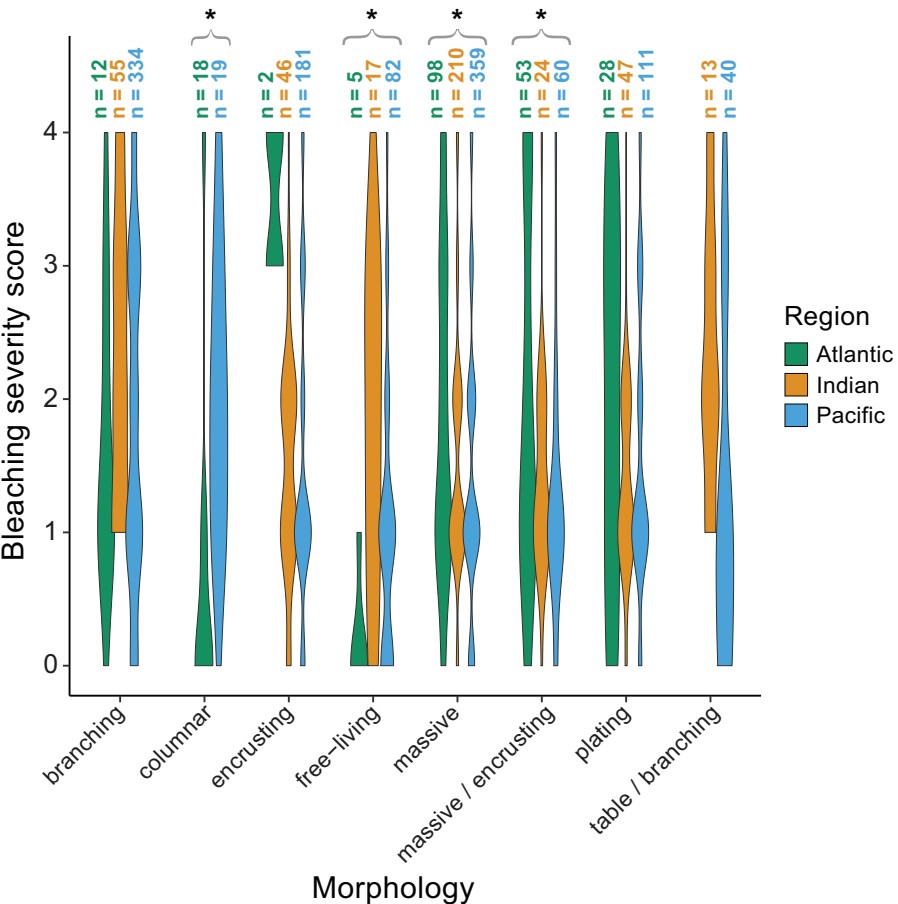

**Figure 6** **Coral bleaching severity by morphology and region.** Violin plots showing the distribution of bleaching severity scores for each common coral morphology across the Atlantic, Indian, and Pacific Oceans (0 = none, 1 = low, 2 = moderate, 3 = high, 4 = severe). The total number of observations by genus and morphology is indicated above each box. Asterisks indicate morphologies for which mean bleaching severity varied significantly by region.

Table S2) involved scoring colonies by their bleaching severity category (0% bleached, 1–20% bleached, 21–50% bleached, *etc.*) and calculating a weighted mean for each taxon representing bleaching index (also known as "BI", Bleaching Response, or Bleaching Mortality Index). Alternatively, rather than categorizing the bleaching state of an entire colony, 4.1% of studies (three out of 74; Table S2) estimated the percentage of surface area within a colony that appeared bleached, although these within-colony observations were much less common. 10.8% of studies (eight out of 74; Table S2) used qualitative observations (*e.g.*, "*Acropora* was more impacted than *Porites*") based on descriptions from the field. One study (*Knipp et al., 2020*) compared the Coral Watch health scores of individual colonies (*via* the color reference card from *Siebeck et al. (2006)*) against SST anomalies at the time of bleaching in order to rank taxa by their "cross-correlation coefficient". To our knowledge, this is the only study to incorporate both regional thermal stress and coral color response data.

58.1% of studies (43 out of 74; Table S3) designated bleaching observations only to coral genus rather than to the species level. Given that coral taxonomy is complicated, arbitrary whether through imagery or *in situ*, and continually changing (*Veron, 2011*), even species-specific studies sometimes grouped certain taxa by genus or species complex (*e.g.*, the *Montastraea annularis* species complex comprising *M. annularis*, *M. faveolata*, and *M. franksi*, which have since been moved to *Orbicella*). In terms of sampling time frame, observations were taken anywhere between immediately at the onset of bleaching (32.4% of studies), to weeks (27.0% of studies) or months (12.2% of studies) later, and up to a year post-bleaching (4.1% of studies; Table S4), although some of these studies also measured recovery or mortality. Only 24.3% of studies (18 out of 74; Table S4) had observations of coral bleaching at multiple time points before, during, or after the mass bleaching event.

## DISCUSSION

The occurrence and magnitude of coral bleaching worldwide is increasing with the progression of climate change (*Baker, Glynn & Riegl, 2008*; *Hughes et al., 2017a*). While the field of coral reef ecology has advanced considerably in the last few decades, bleaching responses are still challenging to predict on a smaller scale (*i.e.,* taxon or colony-level, or even among adjacent conspecifics). Variability in bleaching responses is to be expected since corals have distinct bleaching thresholds related to temperature fluctuations in a given location (*Carilli, Donner & Hartmann, 2012*). Variability could also be linked to contextual environmental factors such as high human influence and proximity to urban areas (*Sandin et al. 2008*; *Smith et al., 2016*), although the relationship between isolation from local stressors and coral reef resilience has been contested (*Baumann et al., 2022*). Further, while corals with a prior history of bleaching may be more acclimatized to warmer temperatures (*Coles et al., 2018*; *DeCarlo et al., 2019*) or exhibit local adaptation (*Barshis, 2015*), climate change is interfering with the potential for coral adaptive mechanisms to support physiological resilience and thermal tolerance (*Ainsworth et al., 2016*). Indeed, cumulative and repeated thermal stress events can turn coral taxa previously deemed 'winners' into 'losers' (*Grottoli et al., 2014*), and responses at the population or community level can be related to the extent and intensity of prior heatwaves (*Fox et al., 2021*; *Wall et al., 2021*). Here, we reviewed the various ways in which observational bleaching measurements are taken and, after standardizing past measurements, we explored whether coral genera and morphologies showed differential bleaching responses among regions.

While coral genera have evolved independently over time and few genera are shared across regions, nor are they equally present within a given region, quantitative comparative studies such as this one can still reveal emerging patterns. Coral reefs are projected to decline in cover on a global scale (*Hughes et al., 2017a*; *Pandolfi et al., 2011*), yet different regions have been, and will likely continue to be, disproportionately affected by climate change impacts (*Shlesinger & Van Woesik, 2023*; *Sully, Hodgson & van Woesik, 2022*). Previously, coral genera were found to have contrasting susceptibilities to bleaching in different sites within Southeast Asia (*Guest et al., 2012*) and this was thought to correspond to each location's thermal history; less severe bleaching was associated with greater historical

temperature variability. Bleaching was also less common near the equator, where thermal variability is generally higher (*McClanahan et al., 2020*; *Sully et al., 2019*). Here, our finding that *Acropora* was less susceptible to bleaching in the Atlantic (Fig. 5) is interesting when considering Caribbean-wide mass declines of *Acropora* since the 1960s due to disease or local anthropogenic stressors (*Cramer et al., 2021*; *Cramer et al., 2020*; *Perry et al., 2015*). However, perhaps following the loss of less-resilient colonies, the remaining Atlantic acroporids are more stress-tolerant (whether through the physiological traits of the symbionts hosted, a post-disturbance shift to more thermally-tolerant symbionts, and/or acclimatization of the holobiont).

Nevertheless, given the inconsistencies in bleaching response metrics (Table S2), it is difficult to standardize results from observational studies on coral bleaching, as discussed by *Grottoli et al. (2021)* for experimental studies. For example, studies have reported bleaching "prevalence" (*e.g.*, *Williams et al., 2010*), "susceptibility" (*Chou et al., 2016*; *Dalton et al., 2020*), "sensitivity" (*Darling, McClanahan & Côté, 2013*), "frequency" (*Montano et al., 2010*), or "severity" (*Guest et al., 2016*), yet these metrics are not necessarily interchangeable since they do not quantify the same aspect of bleaching. Moreover, percent bleaching can refer to either the tissue area that is bleached within a single colony (*Glynn et al., 2001*; *Montano et al., 2010*) or the proportion of total colonies displaying signs of bleaching as opposed to those that were unaffected (*Bruno et al., 2001*; *Carroll, Harrison & Adjeroud, 2017*; *Miller, Piniak & Williams, 2011*). Thus, there is a need to consolidate not only our methods for measuring bleaching, which in themselves range in spatial scale and level of resolution (Fig. 4; Table S1), but also our terminology when reporting bleaching. A robust, consistent metric for bleaching and recovery is critical for correctly determining the factors driving these processes. Given that *in situ* observations are usually taken after the onset of bleaching (Table S4), the timing of surveys with respect to a bleaching event can also confound our interpretations of results (*Claar & Baum, 2019*). Further, most studies do not identify corals to the species level (Table S3) nor track colonies through time (but see: *Ritson-Williams & Gates, 2020*). Species-specific measurements at the individual colony level would minimize some of the uncertainty seen in bleaching responses.

Additionally, since bleaching incidence is more often observed in areas of higher accumulated thermal stress (Fig. 3), it may be necessary to take into account the magnitude and duration of thermal stress when characterizing bleaching responses. Such analyses are for the most part unprecedented (but see: *Knipp et al., 2020*, in which coral coloration was cross-correlated with temperature data) and would lend themselves to more accurate inter-study comparison. We also need to account for prior bleaching history and environmental legacies (*Bahr, Rodgers & Jokiel, 2017*; *Brown et al., 2002*; *Wall et al., 2021*), since thermal adaptation and resilience has been demonstrated in some cases (*Coles et al., 2018*; *Guest et al., 2012*; *Logan et al., 2014*; *Palumbi et al., 2014*). For a more nuanced perspective on bleaching, post-stress recovery, and the underlying factors contributing to resilience and/or resistance at physiological and ecosystem scales, it is important to continue monitoring the same individual coral colonies in the long term (>1 year) where possible.

## CONCLUSIONS

Here, we show that *in situ* coral responses to thermal stress not only vary by region, genus, and morphology but also that some genera or morphologies have unique regional responses to thermal stress. This may be due to different regional stressors and adaptation, the duration of incurred thermal stress, and/or population or community-level differences in symbionts and holobionts. However, inconsistencies in bleaching measurements and reporting between studies can obscure specific findings as well as possible causes of this variability. Standardized, quantitatively-compatible bleaching response metrics that incorporate the severity of thermal stress and other contextual factors (*e.g.*, local stressors) would be more useful for predicting bleaching susceptibility, and will improve comparability across studies. It should be noted that this is inherently complicated since corals exist in different environmental contexts, each with their own thermal history experiencing distinct combinations of local and global stressors, and comparing one reef to another may not always be appropriate. Still, disentangling some of these discrepancies will lead to a better understanding of coral bleaching and recovery dynamics and could help to inform more effective management of coral reefs in the face of climate change.

## ACKNOWLEDGEMENTS

We thank J Shurin and three anonymous reviewers for feedback on this manuscript.

### Funding

Adi Khen was supported by the National Science Foundation (NSF) Graduate Research Fellowship and the Scripps Family Foundation. Chris Wall was supported by NSF Award No. 2018058. The funders had no role in study design, data collection and analysis, decision to publish, or preparation of the manuscript.

### Grant Disclosures

The following grant information was disclosed by the authors:
The National Science Foundation (NSF) Graduate Research Fellowship.
The Scripps Family Foundation.
NSF Award No.: 2018058.

### Competing Interests

The authors declare there are no competing interests.

### Author Contributions

- Adi Khen conceived and designed the experiments, performed the experiments, analyzed the data, prepared figures and/or tables, authored or reviewed drafts of the article, and approved the final draft.
- Christopher B. Wall analyzed the data, authored or reviewed drafts of the article, and approved the final draft.

- Jennifer E. Smith conceived and designed the experiments, authored or reviewed drafts of the article, and approved the final draft.

## Data Availability

The full literature review dataset (including metadata) and data subset compiled for this study as well as R code are available in the Supplementary Files.

## Supplemental Information

Supplemental information for this article can be found online at http://dx.doi.org/10.7717/peerj.16100#supplemental-information.

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
