# Peer review of "Standardization of in situ coral bleaching measurements highlights the variability in responses across genera, morphologies, and regions"

_PeerJ, doi:10.7717/peerj.16100_

## Round 0.1 · original submission · Major Revisions

Dear Dr Khen and co-authors,

I have now received three independent reviews of your study. While all reviewers clearly recognised the quality/novelty of your work, they have collectively raised a number of major issues that will need to be addressed in your revised manuscript.

Overall, the reviewers have provided you with excellent suggestions on how to improve the manuscript, and I be looking forward to receiving your revised manuscript along with a point-by-point response to their comments.

With warm regards,
Xavier

Reviewer 1 ·

Basic reporting

The paper is well-written, and the introduction/background clearly sets the paper into context. Literature is well referenced and relevant throughout, however in many instances I suggest adding an “e.g.” to the references cited (as they are often not the only papers relevant to what is being discussed).

Experimental design

The methodology requires additional detail and clarity (see major comments below).

Validity of the findings

The paper addresses the goals set out in the introduction, and conclusions are well-stated and linked to original research question. Gaps and future directions (in particular, the importance of standardization in coral bleaching research and specifically observational coral bleaching studies) are clearly stated.

Additional comments

Major comments:

1. While the study is useful and well-done, the methodology requires some additional details/clarity:

a. It should be made clear early on that this survey only included papers which quantified coral bleaching response through observation (i.e., not measuring physiology etc.), and those from natural bleaching events in situ (i.e., not from tank experiments). This is done on line 229 by stating that data was compiled from publications on “coral bleaching events and subsequent responses in situ” but should also be made clear in the introduction (I suggest in lines 221-224), and I would also suggest using the words “observational studies” in lines 228-229 for consistency and clarity.
(See also minor comment regarding lines 176-177).
I also think that “observational” should be added to the title, if possible.

b. It should be made clear whether the 62 published scientific studies identified were from a specific range of years or were from ‘all-time’ in Google Scholar and Web of Science. If ‘all-time’ what were the years which they spanned?

c. Can you provide more detail on the geographic regions, reef habitats, depths, and range of sampling times; perhaps just list these in parentheses within the text?

d. 1st paragraph of the Data Collection and Analysis section is hard to follow. Line 249-250: I suggest adding to read “Most commonly within the literature identified, >80%...”. As is, it initially made me think that those were the categories the authors were using. I am not sure what is meant by lines 251-253 – suggest rewording? Ultimately, it remains unclear to me how the bleaching severity categories were converted to the numerical scores; can this be explained in more detail, or be reworded to make it more explicit? Or perhaps a schematic diagram would be useful for the reader to visualize the process (if possible?! – maybe not).

e. Lines 257-258: Were none of the studies identified conducted in the Caribbean, or were these included in the Atlantic Ocean category? Clarity required here. Furthermore, it could be argued that the study from the Red Sea could be included in the Indian Ocean category; is there a reason this had its own category while the Caribbean Sea did not? If so, this should be explicitly stated. However, on line 260 the authors then state “the three main tropical basins” suggesting that the Red Sea was then added into the Indian Ocean… If that is the case, I would suggest stating that 15 were from the Indian Ocean and not mentioning the Red Sea in the previous sentence so as to not add confusion.

Perhaps alter/simplify all to “10 were from the Atlantic Ocean (including the Caribbean Sea), 15 were from the Indian Ocean (including the Red Sea), and 37 were from the Pacific Ocean.”

f. Lines 332-334 and Fig 6: Can you confidently state that encrusting morphologies bleached more severely in the Atlantic compared to the Indian and Pacific when they only had an n of 2?

2. Some work is needed on the general flow of the paper. In the discussion section, the region/genus/morphology results are discussed first, followed by discussion on methodology, whereas in the results section these are presented the other way around. I would keep this consistent in both sections, with the same topic discussed first in both sections.

Minor comments:

Line 46: Should be “…termed “coral bleaching”. …”. (period currently in the wrong place)

Lines 47-48: Many more references can be provided for this statement other than Baird & Marshall. I suggest adding in “e.g.,” and providing a few more example references.

Lines 48-49: “Under prolonged duration (weeks to months) and/or increased magnitude of thermal stress, bleaching can lead to coral mortality (Cook et al. 1990)” – I suggest changing to “This can be exacerbated by prolonged duration (weeks to months) and/or increased magnitude of thermal stress (Cook et al. 1990).”

Lines 70-71: I would explain here how the “local bleaching threshold” is determined (i.e., one degree Celsius above maximum monthly mean).

Lines 71-72: Suggest defining here what is meant by “mass coral bleaching” (i.e., multiple geographic locations experiencing bleaching simultaneously).

Line 83: For the purpose of this study are “global bleaching events” the same as “mass coral bleaching events”? See previous comment on defining meanings.

Line 108: Perhaps Symbiodinieaceae “species” instead of “symbionts”?

Lines 110-113: The wording of this sentence is confusing. Authors state that “…after bleaching, corals may replace their symbionts during bleaching…”. Suggest rewording for clarity of meaning.

Lines 114-117: The wording of this sentence is also confusing – symbionts were depleted post-bleaching? Do you mean that corals with significantly lower symbiont densities post-bleaching? The way it is currently worded makes it sound like the corals didn’t lose their symbionts until after the bleaching event.

Lines 108-129: New and relevant papers on this topic are de Souza 2022 (https://doi.org/10.1098/rsos.212042) and de Souza 2023 (https://doi.org/10.1038/s41598-023-35425-9).

Line 135: I believe this should be “…as well as driving…”.

Lines 137-143: Here, “resistance to thermal stress” is defined, but I believe this should come much earlier on as resistance is discussed prior to this paragraph, too. It seems somewhat out of place in its current position.

Lines 176-177: It could be argued that lab analyses are also used to assess bleaching (i.e., chl-a concentration and endosymbiont cell density). Can this be worked in here? Or can it be made clear that those listed are only visual/observational bleaching assessment methods?

Line 196: Should be “tens-of-meters”.

Lines 207-209: Another thought within this topic is how partially bleached colonies are dealt with. Can a few sentences on this be included?

Line 347: I think the 2017 Hughes papers should be listed in the references the other way around; currently 2017b is cited first (lines 68-69, 157, and 185) and 2017a is cited second (line 347).

Lines 347-349: Worth mentioning here the fact that adjacent colonies of the same species may respond to thermal stress differently (one bleaches while the other doesn’t)?

Lines 368-374: I am not sure the authors should be comparing their results on bleaching severity in relation to region to previous results on bleaching frequency (Hughes et al. 2018), as they are very different… you can have high severity with low frequency and vice versa…

Lines 379-384: I think the impact of the local stressors (i.e., tourism and overfishing) should be highlighted here with regards to the loss of Acroporids in the western Atlantic – see also Cramer et al. 2020 (https://www.doi.org/10.1126/sciadv.aax9395). I also suggest changing “more-susceptible” to “less-resilient” on line 383.

Line 419-420: Not sure what is meant by “incurred heat stress here” – I think it requires more specificity; do you mean duration, or extent, perhaps?

Line 421: Suggest changing to “…inconsistencies in bleaching measurements and reporting between studies…”.

Line 424: Perhaps “…for predicting bleaching susceptibility” would be better here?

Line 425: Not sure what is meant by “different systems” – do you mean “local environments”?

Reviewer 2 ·

Basic reporting

This manuscript reviewed coral bleaching across genera, morphologies, and regions.
The manuscript was written using clear, unambiguous, professional English language throughout. The intro and background show context. The literature was well referenced and relevant, with a few missing references (noted below). The structure conforms to discipline norms and is clear. The review is of broad and cross-disciplinary interest, and is within the scope of the journal.
The field has been partially reviewed in 2018 (Claar et al., 2018, focus specifically on El Nino and coral bleaching), and partially reviewed in 2023 (Baum et al., 2023, focus specifically on local anthropogenic influence and bleaching). Neither of these papers are cited in the current manuscript. I do think that this review provides a different point of view and should be published, but I am concerned that some literature may be missing from the review.
The introduction adequately introduces the subject and makes it clear who the audience is and what the motivation is.

Experimental design

The article content is within the Aims and Scope of the journal. The study was performed to an adequate technical and ethical standard, and methods were described with nearly enough detail and information to replicate but see comments re: literature review search terms below. The Survey Methodology appears to be unbiased, and sources are generally adequately cited and paraphrased as appropriate. The review is organized logically into coherent paragraphs/subsections.

Validity of the findings

Conclusions are well stated, linked to original research question and limited to supporting results. There is a well developed and supported argument that aims to meet the goals set out in the Introduction, although I have a few questions about how the coral bleaching scores were compared among studies (see below). The Conclusion identifies unresolved questions / gaps / future directions.

Additional comments

Major comments
I am concerned that studies may be missing from the review due to potentially missing search terms. Typically wildcards are used during systematic literature searches to ensure that different versions of each word are captured. Please run the literature search again using something like “(coral*) AND (bleach*)” or similar to ensure that you aren’t missing literature. You can also cross check with the recent literature reviews to see if anything is missing. If you’ve already done this, just specify that in Lines 231-232 – currently says “(e.g., “coral bleaching”, “bleaching severity”, “bleaching index”, “mass bleaching”)”. Also include the date that the search was conducted.

The other major question I have is about using relative severity within each study, then translating to 0-4 scale. I am curious how your results might change if you instead standardized to percentages and then to the 0-4 scale. (i.e., all 1s are <20%, all 2s are 20-40%, all 3s are 40-80%, and all 4s are 80%+ [I’m not sure about these exact bins, this is just given as an example]). It seems like this would standardize more across regions rather than comparing local severity. Ideally, this analysis would be included in the manuscript, but at the very least, this should be discussed in the discussion.

Minor comments:
Lines 64-65 – Remove “decreased sea water temperature” and citations here. I agree that this can instigate bleaching, but in the context of this sentence (which starts with warming) it is a bit out of place.
Line 108 – Swap the word “known” for “posited” or similar. This may seem semantic, but there is so much nuance to our understanding of this statement and how it applies to what happens in the field (which you delve into a bit in the rest of the paragraph), that I think this distinction is important.
Line 181 – “only captures emergent reefs and relies on predictions from temperature metrics” I’m not sure what you mean by “only captures emergent reefs” here – recommend to clarify or remove this phrase
Lines 205-207 – image analysis of coral planar areas can also be influenced/potentially biased by coral morphology, since the more rugose a coral is, the more complicated interpretation of planar bleaching area is. (It’s still a very good method, but this caveat should be noted)
Lines 212-213 – how would you define achieving “a sufficient level of taxonomic resolution”? Perhaps reword to say – require extensive training data to achieve a sufficient level of taxonomic resolution – or similar.
Figures:
Check that permissions are in place for adapted figures
Figure 5/6 – These figures could be improved by adding points for each individual study and/or by switching to a violin plot so that distribution is more clearly shown. This may be difficult for Fig 5 given the number of genera, but should be attempted, if possible.

Reviewer 3 ·

Basic reporting

Khen et al. provide an interesting commentary on the need to standardize coral bleaching measurements. While the introduction provides a comprehensive review of the various factors that can contribute to variability in coral bleaching response, certain points made regarding the Adaptive Bleaching Hypothesis (ABH) are outdated. Referring to the ABH as a potential physiological and evolutionary response is misleading and incorrect. Previous papers have extensively discussed this hypothesis over the past two decades, and to date, there is very limited or no data supporting the ABH.

Experimental design

I was surprised to see that only 62 peer-reviewed papers were found to compare bleaching variability. The authors identified several aspects in regard to taxa, morphotype, and regions that influence bleaching prevalence. Overall, the results of this study are somewhat underwhelming, that is follow the same colonies over time and consider the timing and severity of the bleaching episode. I believe that this would indeed help understand individual and community-scale bleaching, however, I do not think it achieves the global standardization metrics needed for cross-comparing studies.

Validity of the findings

While this work raises interesting points and considerations, it mostly addresses causes of variability that are already recognized in the field. To further enhance the understanding of cross-reef or cross-basin variability, a meta-analysis could be conducted. The current figures in the manuscript do not sufficiently explore the variability aspects that are meticulously explained and criticized in the text.

Documenting bleaching is often a subjective process, and it would be helpful to recommend specific techniques to minimize bias. Providing such recommendations could improve the accuracy and reliability of bleaching assessments.

Additional comments

Line 44: Aren’t marine heatwaves and elevated thermal stress the same? If not, please describe the difference.

Line 57: I am not sure that Fitt et al. 1993 is the correct citation for this statement. In fact, "switching" vs. "shuffling" has been discussed for years, however, there is very little evidence to support the uptake of new symbionts from environmental reservoirs i.e., "switching".

Lines 110-114: The ABH provided a theoretical platform to test several hypotheses 3 decades ago. However, it is outdated and isn't supported by the hundreds of studies focused on Symbiodiniaceae changes post-bleaching.

Lines 114-116: These findings (Toller et al. 2001) have been subsequently followed up and there is no strong evidence that these "new" symbionts were taken up from the environment. In fact, the "E" symbiont was identified as Durusdinium trenchii, a symbiont known to occur in low densities and become abundant post-bleaching (See Mieog et al. 2007; Thornhill et al. 2006; Kemp et al. 2014).

This review of switching vs. shuffling is outdated and not fully correct. I suggest not including this as it isn't needed. You can simply state that symbiont assemblages are important in understanding the "bleaching response" of a coral species and colony.

Lines 286-289: I completely agree about the widely inconsistent way of reporting bleaching. For years using “completely bleached, mostly bleached, somewhat bleached, pale….etc” has created confusion among scientists and managers. I would like to see examples (photos?) showing the difference between somewhat bleached and pale. Likely, these are physiologically similar. Regardless, effort should be made to minimize this in future studies. I think this could be an excellent place to construct a figure demonstrating the silliness of terms like these.

Line 321-329. I don’t see the point of comparing genera across ocean basins. These genera have independently evolved over millions of years. Comparing the two Acropora Caribbean species to the +100 Acropora in the Pacific doesn’t seem appropriate.

Lines 379-382: This is not that surprising. In the Caribbean, they associate with Symbiodinium fitti a species with enhanced thermal tolerance. In the Pacific Acropora spp. associates with several different Cladocopium species, many known to be thermally sensitive.

Tables 1-4 do not advance the narrative of this study. I suggest moving to the supplemental section.

---

## Round 0.2 · accepted · Accept

I am pleased to accept this revised manuscript for publication in PeerJ - Congratulations!

I also take the opportunity to thank the reviewers for their valuable contribution in improving this work.

Please note that reviewer#1 has left one minor correction which you may incorporate during the proofs.

With warm regards,
Xavier

Reviewer 1 ·

Basic reporting

No comment.

Experimental design

I am satisfied that the required detail and clarity in the methodology have been addressed.

Validity of the findings

No comment.

Additional comments

I am satisfied that the authors have address all the issues highlighted by the three reviewers and I advise only one minor edit.

Lines 348-349: Authors state “…60-80% as “high,” 40-60% as “moderate,” …”, however there should be further clarity on the numbers here. Was a study stating 60% mortality designated as high or moderate? It should be either 40-59% and 60-80%, OR 40-60%, 61-80%, right? (Also, NB: commas should not within the parentheses in the text).

Reviewer 2 ·

Basic reporting

The article now meets these standards

Experimental design

The article now meets these standards

Validity of the findings

The findings are valid and I have no additional comments

Additional comments

The authors have incorporated all revisions, and I think this article is ready for publication.

Reviewer 3 ·

Basic reporting

I am glad that the authors took the advice of the reviewers and I believe that the manuscript is greatly improved.

Experimental design

No comment.

Validity of the findings

I believe that the changes the authors made help to streamline their intended message.

Additional comments

I believe that the authors did a satisfactory job with their edits. I have no additional comments to suggest.